# Acetylation of Rec8 cohesin complexes regulates reductional chromosome segregation in meiosis

Ziqiang Li[1,2] , Yu Liu[1,2], Andrew W Jones[3], Yoshinori Watanabe[1]

For establishing sister chromatid cohesion and proper chromosome segregation in mitosis in fission yeast, the acetyltransferase Eso1 plays a key role. Eso1 acetylates cohesin complexes, at two conserved lysine residues K105 and K106 of the cohesin subunit Psm3. Although Eso1 also contributes to reductional chromosome segregation in meiosis, the underlying molecular mechanisms have remained elusive. Here, we purified meiosis-specific Rec8 cohesin complexes localized at centromeres and identified a new acetylation at Psm3-K1013, which largely depends on the meiotic kinetochore factor meikin (Moa1). Our molecular genetic analyses indicate that Psm3-K1013 acetylation cooperates with canonical acetylation at Psm3-K105 and K106, and plays a crucial role in establishing reductional chromosome segregation in meiosis.

## Introduction

In mitosis, duplicated chromosomes (sister chromatids) become connected during the S phase through the action of a multisubunit complex called cohesin, which consists of four core subunits: two SMC (structural maintenance of chromosome) family ATPase proteins, Psm1 and Psm3, a kleisin family protein Rad21, and Psc3 (called Smc1, Smc3, Scc1, and Scc3, respectively, in budding yeast) (Tomonaga et al, 2000). The cohesion of sister chromatids is maintained throughout the G2 phase until metaphase when chromosomes are aligned at the spindle equator. At the onset of anaphase, the anaphase-promoting complex–dependent degradation of the securin (Cut2 in fission yeast) allows the release of the protease called separase (Cut1), which cleaves Rad21 and releases sister chromatid cohesion, leading to the separation of sister chromatids (Yanagida, 2000; Nasmyth, 2001; Peters et al, 2008).

During the mitotic cell cycle, an acetyltransferase called Eso1 in fission yeast (corresponding to budding yeast Eco1) has a key role in establishing cohesion in the S phase (Tanaka et al, 2000). Eso1 mainly acetylates two lysine residues, K105 and K106, of Psm3. Mutations of both lysine residues to asparagine or glutamine, which

mimics the acetylated state, can sustain cell viability in the absence of the eso1 gene, which is otherwise an essential gene (Feytout et al, 2011; Kagami et al, 2011). Analogous findings have been originally made in budding yeast (Rolef Ben-Shahar et al, 2008; Unal et al, 2008). Mutation of both lysine residues to nonacetylatable arginine causes mild cohesion defects in fission yeast but lethality in budding yeast. It has been postulated that other acetylation mediated by Eso1 on the cohesin complex may contribute to the regulation of cohesin function in fission yeast (Feytout et al, 2011; Kagami et al, 2011). In both budding and fission yeast, the lethality of eco1 or eso1 acetyltransferase mutants is largely suppressed by the deletion of a cohesin-releasing factor Wpl1, indicating that cohesin acetylation plays an essential role in preventing Wpl1 function.

In meiosis, the Rad21 subunit is replaced on many cohesin complexes by the meiosis-specific version Rec8, which plays a central role in establishing meiosis-specific chromosome segregation to allow the reduction in chromosome number (reductional division) (Watanabe & Nurse, 1999). Whereas Rad21-containing cohesin localizes preferentially to the pericentric regions of centromeres (pericentromeres), meiotic Rec8-containing cohesin localizes in addition to the core centromere, conjoining the two kinetochore-assembling domains and, thus, promoting the mono-orientation of sister kinetochores at meiosis I (Watanabe et al, 2001; Yokobayashi et al, 2003). Mono-orientation of sister kinetochores requires the Rec8 cohesin complexes and the meiosis-specific kinetochore regulator meikin Moa1 (Yokobayashi & Watanabe, 2005; Sakuno et al, 2009).

Although the cohesin acetyltransferase is required for sister chromatid cohesion and chromosome segregation during meiosis in various organisms (Kagami et al, 2011; Singh et al, 2013; Lu et al, 2017; Barton et al, 2022), how acetylation regulates meiotic cohesin complexes, especially at centromeres, is largely unknown. Here, we purified meiotic centromeric cohesin complexes and analyzed their modifications by mass spectrometry. We identified a new acetylation at the conserved lysine residue Psm3-K1013, which largely depends on Moa1. Our genetic analyses indicate that Moa1-dependent acetylation of Psm3 plays a crucial role in setting up mono-orientation of sister kinetochores and reductional chromosome segregation in meiosis.

[1]Science Center for Future Foods, Jiangnan University, Wuxi, China    [2]School of Bioengineering, Jiangnan University, Wuxi, China    [3]Cell Cycle Laboratory, The Francis Crick Institute, London, UK

Correspondence: ywatanabe@jiangnan.edu.cn

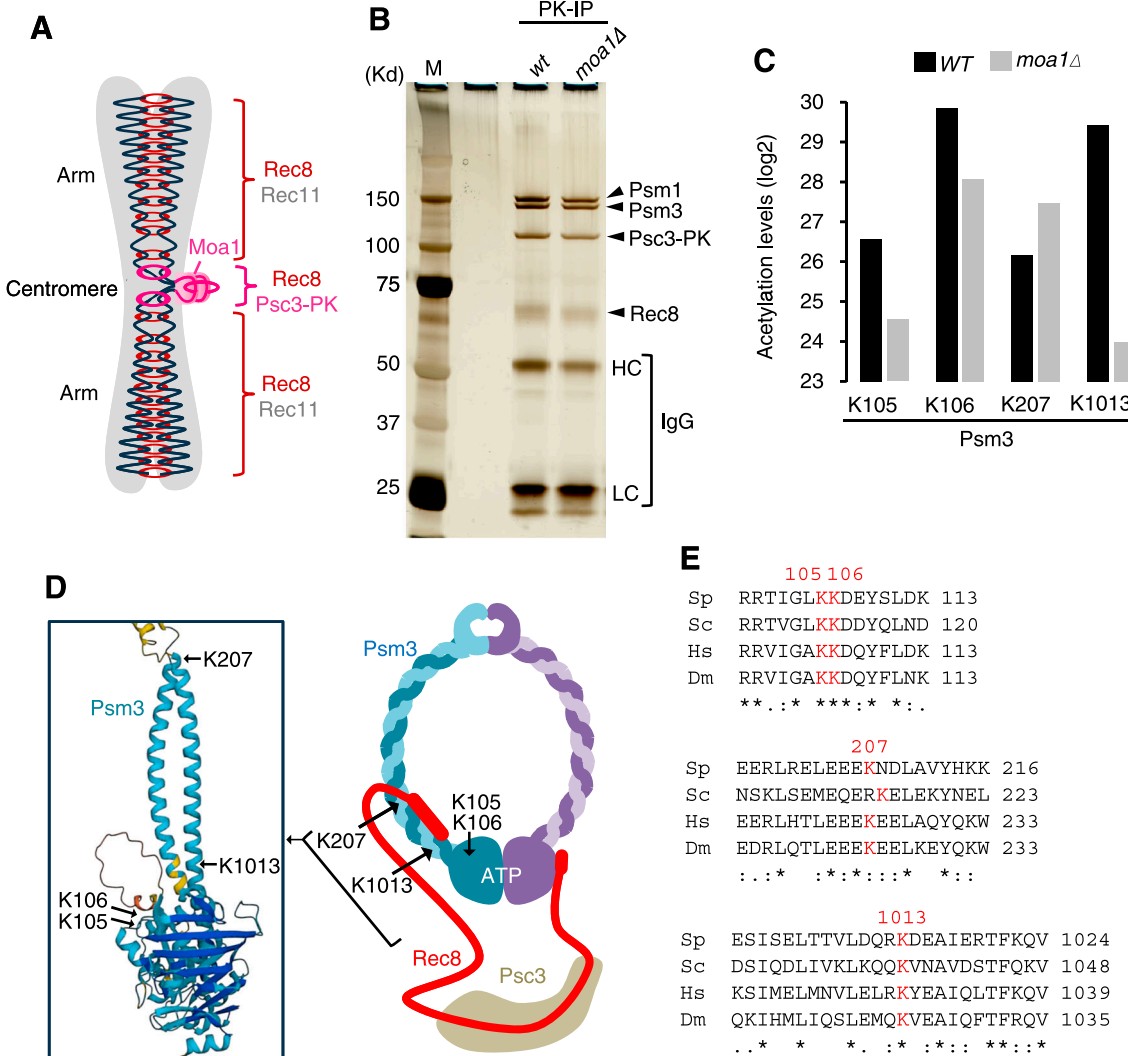

**Figure 1. Identification of Moa1-dependent acetylation in the centromeric Rec8 cohesin complexes.**
**(A)** Schematic depiction of the distribution of the Rec8 cohesin complexes at the centromere and chromosome arms (right). **(B)** Cell extracts were prepared from meiotic cells arrested at prophase I and immunopurified by anti-Pk antibodies. The immunoprecipitates were electrophoresed in an SDS–polyacrylamide gel and visualized by the silver stain. **(C)** Acetylation intensity at the indicated sites of Psm3 detected in the cohesin complexes immunoprecipitated from WT and *moa1Δ* cells (Fig S1A). **(D)** Schematic depiction of the Rec8 cohesin complex and AlphaFold prediction of the Psm3 coiled-coil and head domain (enlarged square). **(E)** Alignment of amino acids around the Psm3 acetylation sites in *S. pombe*, *S. cerevisiae*, *H. sapience*, and *D. melanogaster*.

# Results and Discussion

## Purification of centromeric cohesin complexes from meiotic cells

Our previous studies in fission yeast suggested that mono-orientation defects caused by the deletion of *moa1* are partly suppressed by a mutation in the cohesin deacetylase *clr6*, suggesting that acetylation contributes to the establishment of mono-orientation of sister kinetochores (Kagami et al, 2011). Therefore, we sought to identify acetylations potentially involved in this regulation by analysis of the Rec8 cohesin complexes localizing at the centromeres of meiotic chromosomes. Meiotic cohesin complexes at centromeres contain the Psc3 subunit, which is largely replaced

by its meiotic homolog Rec11 along the chromosome arm regions (Kitajima et al, 2003). Therefore, to exclusively purify centromeric meiotic cohesin complexes, we tagged Psc3 with 3xPK (Fig 1A and Table S1). We prepared cell extracts from WT and *moa1Δ* cells in prophase I of meiosis and performed immunoprecipitation using an anti-PK antibody. All four cohesin subunits Rec8, Psm1, Psm3, and Psc3-3PK were recovered in the immunoprecipitates (Fig 1B). Through analysis by mass spectrometry, we detected several acetylations in all cohesin subunits (Fig S1A and B). Among them, acetylation at Psm3-K1013 was abundant in WT cells, but exceptionally >40 times lower in *moa1Δ* cells (Fig 1C). The site of this acetylation is predicted to be spatially close to the canonical acetylation sites Psm3-K105 and K106 (Fig 1D), acetylation of which

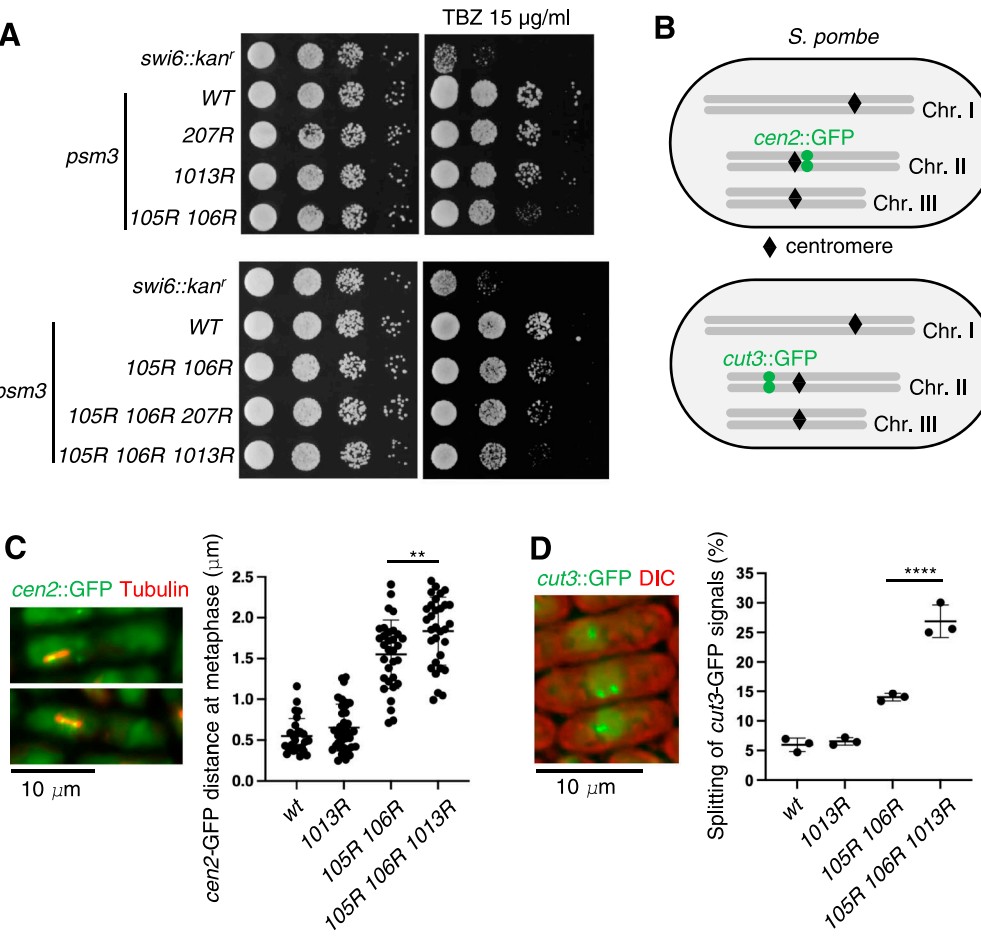

**Figure 2. Acetylation at Psm3-K1013 cooperates with that at K105 and K106 for establishing cohesion in mitosis.**
**(A)** Serial 10-fold dilutions of the indicated cells were spotted on YE plates and grown at 28°C for 2 d (left) and 4 d (right, including TBZ). **(B)** Schematic depiction of *S. pombe* chromosomes marked by GFP. **(C)** Sister chromatid cohesion at metaphase was observed. Distances of sister *cen2*::GFP were measured in the indicated cells, which were arrested at metaphase (by *cut9-665*). 27, 35, 32, and 31 cells were used in the indicated strain, respectively. Error bars represent the SD. **$P < 0.01. **(D)** Sister chromatid cohesion at the arm domain was observed in asynchronous cells (mostly in interphase). Distances of sister *cut3*::GFP were measured in the indicated cells. 230 ~ 360 cells were used in the indicated strain. Error bars represent the SD (three independent experiments). ****$P < 0.0001.

was reduced three or four times in *moa1Δ* cells (Fig 1C). Hereafter, we focus on the Psm3-K1013 residue in addition to the canonical acetylation sites Psm3-K105 and K106. As a reference, we also analyzed the Psm3-K207 residue, which is acetylated in WT cells and whose acetylation is not reduced in *moa1Δ* cells (Fig 1C). Both the Psm3-K207 and Psm3-K1013 residues seem as conserved as the canonical acetylation sites, Psm3-K105 and K106 (Fig 1E).

### Acetylation at Psm3-K1013 cooperates with that at K105 and K106 for establishing cohesion in mitosis

We made *psm3-K1013R* and *psm3-K207R* cells, in which the acetylation sites are mutated to nonacetylatable arginine, and examined their vegetative growth in the presence of TBZ (thiabendazole), a microtubule-depolymerizing drug. We found that *psm3-K105R/K106R* cells showed a mild sensitivity to TBZ, but neither *psm3-K207R* nor *psm3-K1013R* cells showed TBZ sensitivity (Fig 2A, top). Notably, *psm3-K105R/K106R/K1013R* cells showed a mild enhancement of TBZ sensitivity compared with *psm3-K105R/K106R* cells, whereas *psm3-K105R/K106R/K207R* cells were equally sensitive as *psm3-K105R/K106R* cells (Fig 2A, bottom). To directly detect sister chromatid cohesion defects in mitosis, we monitored GFP-LacI fluorescence associated with a *lacO* array integrated at the centromere of chromosome II (*cen2*::GFP) using metaphase-

arrested cells (Fig 2B). Similarly, cohesion at the arm domains was assayed using cells marked with *cut3*::GFP (GFP-LacI bound to a *lacO* array integrated next to the *cut3⁺* gene on chromosome II) (Fig 2B). In both assays, cohesion defects are mild in *psm3-K105R/K106R* cells and enhanced in *psm3-K105R/K106R/K1013R* cells (Fig 2C and D). We conclude that acetylation at K105/K106 and K1013 cooperates to establish sister chromatid cohesion at the centromeres and chromosome arms in proliferating cells.

### Acetylation at Psm3-K1013 promotes the mono-orientation of kinetochore in meiosis

To examine the requirement of cohesin acetylation for mono-orientation in meiosis, we analyzed meiotic chromosome segregation in the acetylation-defective mutants. We marked *imr1*::GFP on only one of the two homologous chromosomes and monitored its segregation during meiosis I (Fig 3A and B). Although *psm3-K105R/K106R* cells showed defects in mono-orientation (34.6% equational segregation), *psm3-K1013R* or *psm3-K207R* cells showed no defects (Fig 3C). However, *psm3-K1013R* remarkably enhanced the defects of *psm3-K105R/K106R* from 34.6% to 74.3%, whereas *psm3-K207R* did not. We conclude that acetylation at Psm3-K1013 together with canonical acetylation at Psm3-K105/K106 plays a crucial role in establishing mono-orientation of kinetochores in

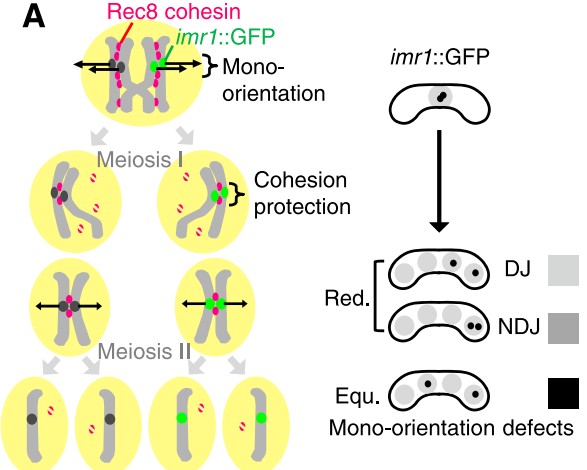 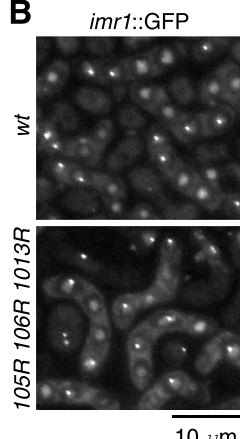

**Figure 3. Acetylation at Psm3-K105, K106, and K1013 cooperatively acts to establish cohesion in meiotic chromosomes.**
**(A)** Schematic depiction of behaviors of homologous chromosomes and Rec8 cohesin during meiosis, showing *imr1*::GFP marked in one homolog. DJ, disjunction; NDJ, nondisjunction. **(B)** Representative fluorescent images (maximum z-dimension projections) of meiotic WT and *psm3-K105R/K106R/K1013R* cells. **(C)** Segregation pattern of *imr1*::GFP marked on one homolog was monitored in the indicated cells after the meiosis II division. (n) Cell number used for the assay. Error bars represent the SD. n.s., not significant; ***$P < 0.005$. **(C, D)** Segregation pattern of *imr1*::GFP marked on one homolog was monitored in indicated cells after the meiosis II division as in (C). (n) Cell number used for the assay. Error bars represent the SD. n.s., not significant; **$P < 0.01$.

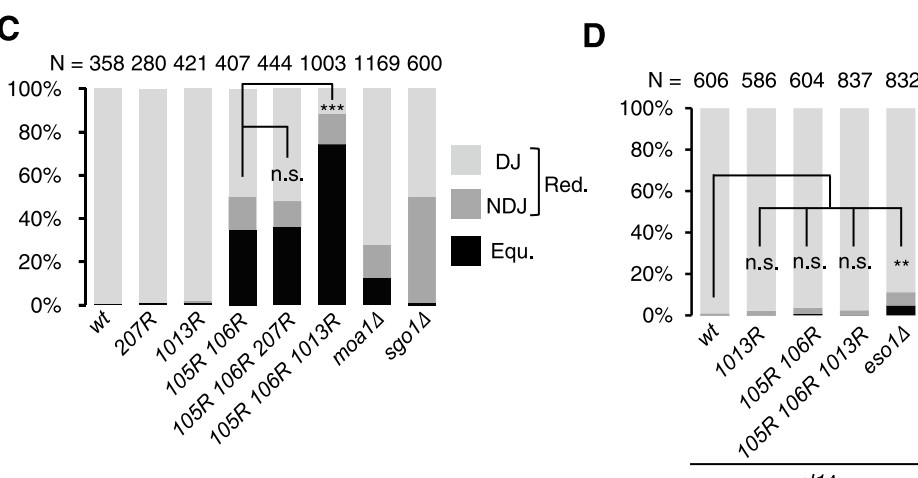

meiosis. Notably, the mono-orientation defects (or equational segregation) in *psm3-K105R/K106R/K1013R* cells were even higher (74.3%) than in *moa1Δ* cells (12.1%) (Miyazaki et al, 2017). If chiasmata are absent (by abolishing recombination through *rec12Δ*), *moa1Δ* cells show >80% equational segregation (Yokobayashi & Watanabe, 2005). However, *psm3-K105R/K106R/K1013R* cells retained substantial recombination (Fig S2A and B), implying that the strong defects in mono-orientation cannot be solely attributed to a lack of chiasmata. Moreover, for the homologs that segregated reductionally, ~50% showed nondisjunction in meiosis II in *psm3-K105R/K106R/K1013R* cells, consistent with random segregation (Fig 3C). This suggests that *psm3-K105R/K106R/K1013R* cells are defective in pericentric cohesion (or cohesion protection) in addition to core centromeric cohesion. *Sgo1Δ* cells, defective in cohesion protection, also show ~50% nondisjunction in meiosis II but no equational segregation in meiosis I (Kitajima et al, 2004) (Fig 3C). These results suggest that cohesion in *psm3-K105R/K106R/K1013R* cells is largely reduced not only at the core centromeres but also at the pericentromeres, thus interfering with proper segregation in meiosis I and II. The defects of reductional chromosome segregation of *psm3-K105R/K106R* and *psm3-K105R/K106R/K1013R* cells were

suppressed by *wpl1Δ* (Fig 3D), indicating that acetylation of these sites prevents the function of the cohesion-releasing factor Wpl1. Although the *KR* mutant phenotype might be possibly due to the shortened side chain rather than a lack of acetylation, we assume that the major phenotype is due to the nonacetylation effect because *psm3-K105R/K106R/K1013R* and *eso1Δ* are similarly suppressed by *wpl1Δ* (Fig 3D).

### Acetylation at Psm3-K1013 promotes cohesion along the whole chromosomal regions

To delineate the requirement of Psm3 acetylation at K105, K106, and K1013 for sister chromatid cohesion in meiosis, we sought to directly examine sister chromatid cohesion using GFP. We monitored *imr1*::GFP at the centromeres of chromosome I and *cut3*::GFP at the arm region of chromosome II (Fig 4A). We used a mutation of *mei4*[+] to synchronize the meiotic cell cycle at late prophase I when sister chromatid cohesion is intact in WT cells (Yokobayashi et al, 2003). Because both homologs are marked by *imr1*::GFP or *cut3*::GFP, cohesion loss can be detected by the emergence of three or four dots (Fig 4A). In WT cells, the *imr1*::GFP dots were mostly double or single,

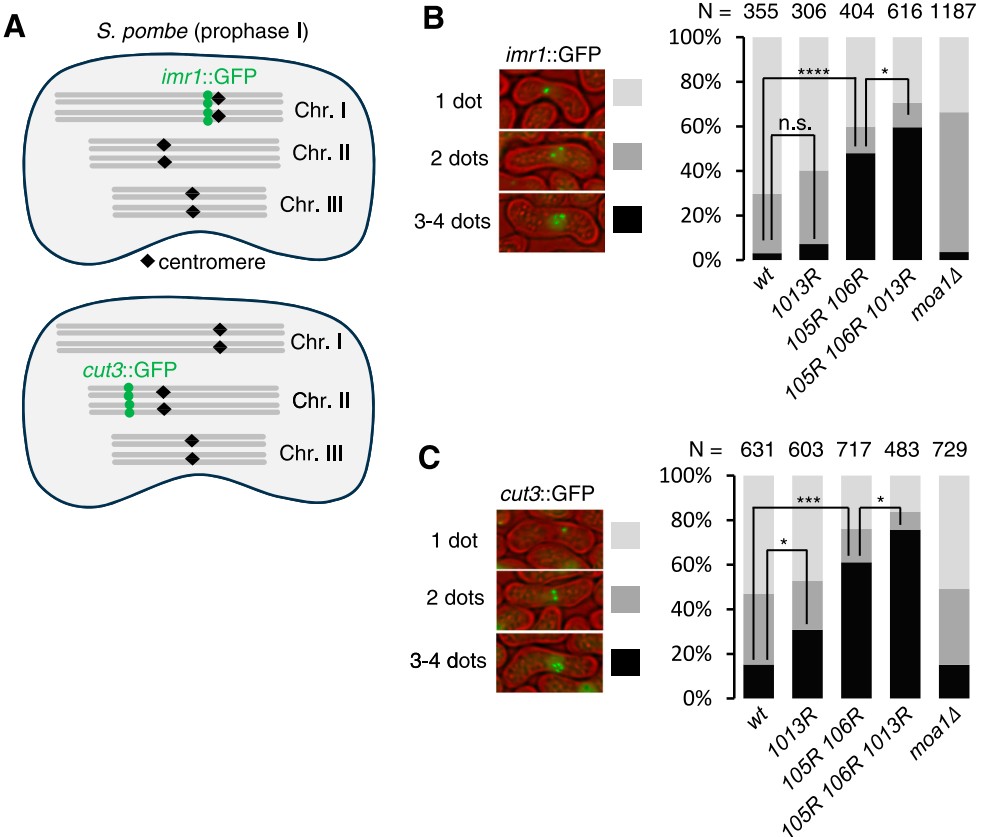

**Figure 4. Psm3 acetylation might be regulated by Moa1 only at the core centromeres.**
**(A)** Schematic depiction of meiotic chromosomes marked by GFP. Note that homologs are paired in meiotic prophase I. **(B, C)** *imr1*::GFP and *cut3*::GFP were observed in prophase I (arrested by *mei4Δ*) $h^{90}$ cells. The number of dots per nucleus is shown with images of representative nuclei. (n) Cell number used for the assay. Error bars represent the SD. n.s., not significant; *$P$ < 0.05, ***$P$ < 0.005, ****$P$ < 0.0001.

indicating that sister chromatid cohesion at the centromere was intact. 70.1% of cells showed one dot, indicating that homologs were paired (Fig 4B). In contrast, three or four dots were detected in nearly 50% of *psm3-K105R/K106R* cells. This suggests a defect of centromeric cohesion, which was further enhanced by the *K1013R* mutation (Fig 4B). Furthermore, cohesion at the chromosome arm was impaired in *psm3-K105R/K106R* cells and even more so in *psm3-K105R/K106R/K1013R* cells (Fig 4C). In contrast, *moa1Δ* cells showed no defects in cohesion at the pericentromere and arm region (Fig 4B and C), consistent with the notion that Moa1 contributes to establishing cohesion only around the core centromere (Yokobayashi & Watanabe, 2005; Sakuno et al, 2009). Notably, centromere pairing is somewhat reduced in *moa1Δ* cells (Fig 4B), possibly by the change of microtubule dynamics at kinetochores, which might be also regulated by Moa1-associated Plo1 (Kakui et al, 2013). These results explain why the defects in mono-orientation (reductional division) are much more extensive in *psm3-K105R/K106R/K1013R* cells than in *moa1Δ* cells; namely, in *psm3-K105R/K106R/K1013R* cells, cohesion defects along all chromosomal regions contribute to the disruption of reductional chromosome segregation, causing equational segregation at meiosis I, similar to *rec8Δ* cells rather than *moa1Δ* cells (Yokobayashi & Watanabe, 2005) (Fig 5).

Because acetylation of centromeric cohesin complexes at Psm3-K105, K106, and K1013 is lower in *moa1Δ* cells (Fig 1B), we reason that Moa1-dependent acetylation at these sites at least partly

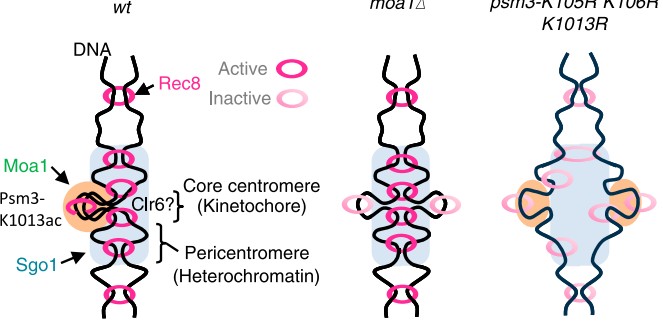

**Figure 5. Schematic model for Moa1 action at the kinetochore in meiosis I.**
Moa1 plays a role in enhancing Psm3 acetylation at K1013, which is important for cohesion establishment at the core centromere and mono-orientation of kinetochores. Cohesin deacetylase Clr6 may antagonize this function because *moa1Δ* is partly suppressed by the *clr6-1* mutation (Kagami et al, 2011).

contributes to the establishment and/or maintenance of cohesion at the core centromere. We previously showed that the deacetylase mutant *clr6-1* partly suppresses mono-orientation defects in *moa1Δ* cells (Kagami et al, 2011). Importantly, *moa1Δ* cells show cohesion defects only at the core centromere but not along chromosome arms (Yokobayashi & Watanabe, 2005). Therefore, a possible scenario is that Clr6 activity might be high only at the core centromere and Moa1 would antagonize the Clr6 activity to promote core centromeric cohesion and mono-orientation (Fig 5). The

defects of cohesion along the chromosome arms in *psm3-K105R/ K106R/K1013R* cells but not *moa1Δ* cells indicate that acetylation at Psm3-K105/K106/K1013 on chromosome arms might be not so much influenced by Clr6 or Moa1. We thus propose that Moa1 directly or indirectly antagonizes Clr6 at the core centromeres, possibly by enhancing the activity of Eso1 or inhibiting Clr6 (Fig 5). Finally, because Moa1-associated Plo1 plays a key role in establishing mono-orientation (Kim et al, 2015; Ma et al, 2021), Plo1 might indirectly regulate this acetylation pathway. Further studies are required to address these hypotheses. Remarkably, after a long-standing debate about the contribution of cohesin to establishing mono-orientation in budding yeast (Toth et al, 2000; Watanabe, 2006; Monje-Casas et al, 2007), a recent study suggests that budding yeast Eco1 and cohesin play an important role in the establishment of mono-orientation of meiotic kinetochores (Barton et al, 2022). Thus, our study together with others further raises the possibility that the acetylation-mediated mono-orientation mechanism might be conserved among eukaryotes, including possibly humans.

# Materials and Methods

### *Schizosaccharomyces pombe* strains and media

The deletion of *moa1+*, *sgo1+*, *mei4+*, *swi6+*, *rec8+*, *eso1+*, and *wpl1+* was performed by the PCR-based gene targeting method for *S. pombe* using kanMX6 (*kan^r*), hphMX6 (*hyg^r*), and natMX6 (*nat^r*) genes as selection markers. To introduce mutations in the *psm3+* gene, we cloned N-terminus and C-terminus fragments of the *psm3* gene into plasmids carrying *nat^r* and *hyg^r* markers, respectively, and they were mutagenized by PCR-based site-directed mutagenesis (Bahler et al, 1998). The mutagenized fragments were integrated in the genome by yeast transformation. Correct mutagenesis was confirmed by PCR and/or DNA sequencing. We used *imr1*::GFP, *cen2*:: GFP, and *cut3*::GFP markers to monitor the cohesion of sister chromatids (Tomonaga et al, 2000; Sakuno et al, 2009). All media and growth conditions unless otherwise stated were as described previously (Moreno et al, 1991). Complete medium yeast extract (YE) or minimal medium (EMM) was used for the culture of *Schizosaccharomyces pombe* strains. A sporulation agar plate was used to induce meiosis. All *S. pombe* strains used are listed in Table S1.

### Immunopurification of Psc3-PK protein complexes from meiotic prophase cells

The temperature-sensitive *pat1-114* (*moa1+* and *moa1Δ*) cells were cultured in 6 liter EMM+N liquid at 25°C (OD$_{660}$ = 0.5). After collecting cells by filtration, cells were resuspended in 4 liter EMM liquid lacking NH$_4$Cl (EMM−N) at a density of OD$_{660}$ = 0.8 and incubated for 7 h at 25°C to induce G1 arrest. Then, after adding 2 ml EMM+N liquid, the culture was shifted to 34°C to induce meiosis in a synchronous fashion. 4 h after the temperature shift, 0.1 mM PMSF was added and mixed, and then, cells were harvested by centrifugation. After washing with 10 ml H buffer (KCl 75 mM, NaCl 75 mM, complete protease inhibitor cocktail [Roche],

PhosSTOP), the pellet was suspended in a half volume of H buffer. The dense suspension was dropped in liquid nitrogen. The frozen cell drops milled by a grinder in liquid nitrogen became powder. 65 ml powder (20 g) was suspended in 20 ml Hb buffer (25 mM MOPS, pH 7.2, 15 mM MgCl$_2$, 15 mM EGTA, 60 mM b-glycerophosphate, 0.1 mM Na-orthovanadate, 0.1 mM NaF, 15 mM *p*-nitrophenylphosphate, 1% Triton X-100, 1 mM dithiothreitol, 1 mM PMSF, complete protease inhibitor cocktail [Roche]) supplemented with Benzonase 10 $\mu$l (2,500 $\mu$). The suspension was sonicated on ice for 20 s twice and centrifuged (A27-8 × 50 rotor, 25,000 rpm × 20 min; Thermo Fisher Scientific). The supernatant was mixed with 2 ml Protein G Magnetic Beads and 40 $\mu$l PK-Ab, and rotated at 4°C for 2.5 h. The beads were washed three times with 2 ml HB buffer and after changing the tube further washed three times. The immunoprecipitates were suspended in 150 $\mu$l SDS loading buffer. A 2 $\mu$l aliquot was applied to SDS–PAGE, followed by silver staining to detect the proteins. Other samples were applied to SDS–PAGE, and the protein bands were detected by Coomassie Brilliant Blue and were cut out for further analysis by mass spectrometry.

### Analysis by mass spectrometry

Protein bands were excised and in-gel–digested using trypsin. The peptides were analyzed with an Orbitrap Fusion Lumos mass spectrometer coupled to an UltiMate 3000 HPLC equipped with an EASY-Spray nanosource (Thermo Fisher Scientific). Raw data were processed using MaxQuant v1.6.0.1 and searched against a UniProt-extracted *S. pombe* FASTA file amended to include common contaminants, with phosphorylation (STY), acetylation (K), and methylation (KR) being selected as variable modifications. The proteingroup.txt and phosphoSTY.txt output tables were imported into Perseus software for further processing. All intensity values were log$_2$-transformed.

### Synchronous cultures of fission yeast meiotic cells

For meiosis microscopic observation, logarithmically growing cells were collected and suspended in 20 mg/ml leucine, spotted on a sporulation agar plate, and incubated at 28°C. For a chromosome segregation assay, *imr1*::GFP was observed in the *mes1+* or *mes1-B44* mutant that arrests at prophase II (after 16 h). For a sister chromatid cohesion assay, *imr1*::GFP and *cut3*::GFP were observed in the *mei4Δ* mutant that arrests at prophase I (after 24 h).

### Sister chromatid cohesion assay of fission yeast mitotic cells

The cells with *cut3*::GFP were cultured in EMM+leucine liquid (OD$_{660}$ = 0.4 ~ 0.6). The number of cells having two *cut3*::GFP signals in a single nucleus was determined. For a centromeric cohesion assay, we used *cut9-665* mutant cells that were cultured at 28°C and shifted to 36°C for 4 h to arrest the cell cycle at metaphase. To visualize tubulin, a mCherry-tagged *atb2+* gene under the *adh13* promoter was integrated at the Z locus (Sakuno et al, 2009). The in-focus fluorescent images were obtained with NIS-Elements software (Nikon), and the distance between two *cen2*::GFP signals on the metaphase spindle (spindle length < 3 $\mu$m) was measured by ImageJ software.

### Recombination assay

Random spores spread on a YE plate (~200 colonies/plate) were incubated at 28°C for 3 d. The colonies were replicated to EMM (+Ura and Leu) and EMM (+Lys and Leu) plates. The number of colonies that can and cannot grow on both plates and the colonies that can grow on one plate and not on the other was counted.

### Fluorescence microscopy

All cell fluorescence microscopy was performed using a Nikon ECLIPSE Ti2-E inverted microscope with a Photometrics PRIME 95B camera. The microscope was controlled by NIS-Elements software. Thirteen z sections (spaced by 0.4 $\mu m$ each) of the fluorescent images were converted into a single two-dimensional image by maximum intensity projection. ImageJ software (NIH) was used to adjust brightness and contrast, and to render maximum projection images.

### Serial dilution analyses

For serial dilution plating assays, 10-fold dilutions of a mid-log phase culture ($OD_{660}$ = 0.4 ~ 0.6) were plated on the indicated medium and incubated at 28°C.

### Statistical analysis

All the data replicates were applied and analyzed using GraphPad Prism version 9.5.1 (GraphPad Software). To estimate the significant differences, we used one-way ANOVA with Bonferroni's multiple comparison test.

# Supplementary Information

# Acknowledgements

We thank Silke Hauf for critically reading the article, Jian Chen for general support, and Yuhei Goto and the Yeast Genetic Resource Center (YGRC) for yeast strains and plasmids. Y Watanabe especially thanks Paul Nurse for providing an opportunity to experiment at the Francis Crick Institute and Takafumi Ishihara for providing a method of immunoprecipitation mass spectrometry. This work was supported by the National Key Research and Development Program of China (2017YFC1600403), the National Natural Science Foundation of China (Key Program, 31830068), and the Entrepreneurial and Innovative Talent of Jiangsu China (JSSCRC2021495).

## Author Contributions

Z Li: data curation, formal analysis, and investigation.
Y Liu: formal analysis and methodology.
AW Jones: data curation, formal analysis, investigation, and methodology.

Y Watanabe: conceptualization, formal analysis, supervision, funding acquisition, project administration, and writing—original draft, review, and editing.

## Conflict of Interest Statement

The authors declare that they have no conflict of interest.

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
