## [Reviewer comments · Life Science Alliance]

Life Science Alliance

Acetylation of Rec8 cohesin complexes regulates reductional chromosome segregation in meiosis

Zigiang Li, Yu Liu, Andrew Jones, and Yoshinori Watanabe

DOI: <https://doi.org/10.26508/lsa.202402606>

Corresponding author(s): Yoshinori Watanabe, Jiangnan University

Review Timeline:

Submission Date:	2024-01-19
Editorial Decision:	2024-02-26
Revision Received:	2024-03-04
Editorial Decision:	2024-03-27
Revision Received:	2024-03-28
Accepted:	2024-03-28

Transaction Report:

February 26, 2024

Re: Life Science Alliance manuscript #LSA-2024-02606-T

Prof. Yoshinori Watanabe
Jiangnan University
Science Center for Future Foods
1800 Lihu Avenue
Wuxi, Jiangsu 214122
China

Dear Dr. Watanabe,

Thank you for submitting your manuscript entitled "Acetylation of Rec8 cohesin complexes regulates reductional division in meiosis" to Life Science Alliance. The manuscript was assessed by expert reviewers, whose comments are appended to this letter. We invite you to submit a revised manuscript addressing the Reviewer comments.

Thank you for this interesting contribution to Life Science Alliance. We are looking forward to receiving your revised manuscript.

Sincerely,

B. MANUSCRIPT ORGANIZATION AND FORMATTING:

Reviewer #1 (Comments to the Authors (Required)):

Li and coworkers have discovered a new acetyl modification of the Smc3 subunit of cohesin, at the conserved lysine 1013 residue. It was identified as an acetyl-mark that is substantially reduced in moa1 mutants, which are defective in meiotic kinetochore mono-orientation. However, further analysis suggests that this acetyl mark has similar importance during meiosis and mitosis. This importance on its own seems to be limited, as unmodifiable K1013R mutants display no defects on their own, but exacerbate the mitotic and meiotic defects of the well-known K105R K106R mutants. This is consistent with a supporting function of K1013 acetylation in cohesin function.

The work in this paper is straightforward, and it will be of interest to the chromosome biology and meiosis community. I have the following comments:

1. One assumes that K1013 acetylation is Eco1 (Eso1)-dependent, but this is never shown or tested. This demonstration (which would require mass-spec analysis) is not critical to the paper, but one wonders if a K1013Q mutant might partially suppress the defects of K105R K106R mutants. Again, not necessary, but the paper would be strengthened.
2. Throughout, the K1013R mutant is conflated with the nonacetylated state, which is true, but it is possible that phenotypes of this mutant are due to the shortened side chain rather than a lack of acetylation. This should at least be acknowledged in the text.
3. Headings that contain "Acetylation at Psm3-K1013 is require for..." show be modified, because, since mutants K1013R single mutants have no discernable defects-thus refuting the assertion of requirement. I suggest "acetylation at Psm3-K1013 promotes..." as a more accurate alternative.
4. The statement that recombination is largely intact in psm3-K1015R,K106R mutants is misleading-there are clear defects. I suggest "psm3-K105R,K106R mutants retained substantial recombination" as alternative wording.
5. Figure EV2 needs a Y-axis label.

Reviewer #2 (Comments to the Authors (Required)):

LSA-2024-02606-T

The paper by Li et al described the characterization of non-canonical lysine acetylation of fission yeast Smc3 (Psm3) protein, a component of the cohesin which is critical for sister chromatid cohesion as well as chromosome-axis formation in both mitotic and meiotic cells. The authors identified the acetylation of K1013 and K207 as well as known K105 and K106 in meiosis-specific cohesin complex with Rec8 as alpha-kleisin and Psc3. By acetylation-defective psm3 mutants, the authors nicely showed that K1013 acetylation plays a redundant role with K105/K106 canonical double-acetylation in chromosome segregation in meiosis I such as monopolar segregation and sister chromatid cohesions (at peri-centromere). The experiments were performed properly and most of the results are technically solid. However, the paper does not provide a molecular insight on the how the loss of Psm3 acetylation leads to cohesin defects in meiosis. It is essential to provide more experimental results to show the molecular mechanism: first, how Moa1 affects the acetylation of Psm3; second, how acetylation of Psm3 promotes mono-orientation of sister-kinetochores; third, how Rec8-cohesin is dissociated from meiotic chromosomes.

Please show page numbers and line numbers. Without these numbers, it is very hard to comment this paper.

Major points:

1. The authors analyzed the role of Psm3 acetylation indirectly by looking at the chromosome segregation. More importantly, the authors have to show the localization of Rec8-cohesin by immuno-localization of Rec8 protein and chromatin immunoprecipitation of Rec8 at various loci including centromere and sub-centromere (and arm) loci.
2. Given the function of Moa1 in Psm3 acetylation, it is important to check the localization of Moa1 in various acetylation-defective psm3 mutants.
3. To know the mechanism of cohesion loss in late prophase I-arrested cells (shown in Figure 4), it is important to know the

effect of the wpl1 mutation in the cohesion loss in various acetylation-defective mutants.

4. Figure 3C and D, Figure 4B and C, graphs: Please show the P-values for the statistical comparison and describe the method in the legend. Without the statistical comparison, it is hard to draw the conclusion.

Minor points:

1. To check the acetylation-defective mutations did not affect the stability of Psm3 protein (and also Rec8), it is very important to analyze the protein level in meiotic cells by western blotting.
2. Figure 1B: This schematic Figure is misleading since the cell used in the two experiments (in Figures 1C and D) has only one GFP marker on the chromosome 2 (rather than two GFP loci on the single chromosomes). The misleading schematic figure is seen in Figure 4A, in which a single yeast cell has two GFP loci in different chromosome: The results in Figures 4B and C were from a cell with single-GFP locus.
3. Figure 1C: How the authors quantify the level of acetylation? How did the authors normalize the values of two strains? What does Y-axis mean? What is the reproducibility of this result?
4. Figure 1E: Please align the amino acid sequences as well the number and marks properly.
5. Figure 2C and D, graphs: Please show the raw data by the dot plot, rather than the bar graph.
6. Figure 4C (and D): In Figure 4D, the result for only the eso1 wpl1 double mutant is shown without that for the eso1 single mutant.
7. Supplemental table: What kind of information in it? Please provide more words to explain the number here.

Reviewer #3 (Comments to the Authors (Required)):

This manuscript investigates the regulation of cohesin acetylation and its role in chromosome segregation during meiosis using fission yeast as a model. The authors first biochemically identify the acetylation of Psm3 within the Rec8 cohesin complex. Some of these acetylations are dependent on Moa1, a meiosis-specific centromere protein required for sister kinetochore mono-orientation in meiosis. They then show that these Psm3 acetylations contribute to chromosome cohesion in general during mitosis. Furthermore, Psm3 mutants at their acetylation sites show defects in mono-orientation during meiosis, similar to moa1D cells, suggesting that these acetylations are necessary for mono-orientation. However, the Psm3 acetylation site mutants show more severe mono-orientation defects than moa1D mutants, more resembling the phenotype of rec8D cells. Combining these results with their previous findings that the deacetylase clr6 mutant partially suppresses the moa1D phenotype, the authors suggest that the role of Moa1 is to promote centromere cohesion by antagonizing the deacetylase Clr6 at the centromere, thereby facilitating mono-orientation.

Overall, this manuscript is well constructed with robust experiments and high quality data, and presents logical and clear arguments. The conclusions are strongly supported by the data and there is no overstatement. Their findings include important advancement in molecular understanding of how meiosis-specific chromosome segregation is regulated. Below are some minor suggestions for possible improvements. I recommend this manuscript for publication.

- I agree with the authors that Moa1 antagonizes Clr6 at the centromere. However, a question remains as to how Clr6 promotes cohesin inactivation at the centromere more than at the arms in the moa1D mutant (this is the impression I got from the model shown in Fig. 5). Is the activity of Clr6 itself higher at the centromere than at the arms? Or does Clr6 have uniform activity along chromosomes, but during meiosis, Moa1 specifically excludes Clr6's action from the centromere? It would be beneficial if the authors could discuss these possibilities.

- Figure legend for Fig. 5 is missing.

Response to the referees,

We thank the referees for their valuable comments. To address the referees' comments, we have revised our manuscript. We addressed all comments raised by the referees and our responses are listed below.

(Bold letters are our responses)

Reviewer #1 (Comments to the Authors):

Li and coworkers have discovered a new acetyl modification of the Smc3 subunit of cohesin, at the conserved lysine 1013 residue. It was identified as an acetyl-mark that is substantially reduced in *moa1* mutants, which are defective in meiotic kinetochore mono-orientation. However, further analysis suggests that this acetyl mark has similar importance during meiosis and mitosis. This importance on its own seems to be limited, as unmodifiable K1013R mutants display no defects on their own, but exacerbate the mitotic and meiotic defects of the well-known K105R K106R mutants. This is consistent with a supporting function of K1013 acetylation in cohesin function.

The work in this paper is straightforward, and it will be of interest to the chromosome biology and meiosis community. I have the following comments:

1. One assumes that K1013 acetylation is Eco1 (Eso1)-dependent, but this is never shown or tested. This demonstration (which would require mass-spec analysis) is not critical to the paper, but one wonders if a K1013Q mutant might partially suppress the defects of K105R K106R mutants. Again, not necessary, but the paper would be strengthened.

***psm3-K1013R* remarkably enhanced the meiotic defects of *psm3-K105R/K106R* from 34.6 % to 74.3%. We indeed examined *psm3-K105R K106R K1013Q* and the defects was 52.1%. A recent report in budding yeast made clear that *KQ* mutant (*smc3-K112Q K113Q*) cells show large defects in cohesin loading although this mutation suppresses lethality of *eso1Δ* (Kaushik et al. *Nature com.* 2023). According to the report, fission yeast *psm3-K105Q K106Q* cells indeed show a mild defect in cohesion (Kagami et al. *EMBO rep* 2011, Fig S1E, see below). Thus, we are reluctant to discuss about *KQ* suppression analysis because the interpretation of the results is complicated and would not strengthen our conclusion.**

2. Throughout, the K1013R mutant is conflated with the nonacetylated state, which is true, but it is possible that phenotypes of this mutant are due to the shortened side chain rather than a lack of acetylation. This should at least be acknowledged in the text.

According to the suggestion, we describe in the text line 140-142; “Although the *KR* mutant

phenotype might be possibly due to the shortened side chain rather than a lack of acetylation, we assume that the major phenotype is due to nonacetylation effect because *psm3-K105R/K106R/K1013R* and *eso1Δ* are similarly suppressed by *wpl1Δ* (Fig 3D)".

3. Headings that contain "Acetylation at Psm3-K1013 is require for..." show be modified, because, since mutants K1013R single mutants have no discernable defects-thus refuting the assertion of requirement. I suggest "acetylation at Psm3-K1013 promotes..." as a more accurate alternative.

According to the suggestion, we changed the heading (lines 115 and 144).

4. The statement that recombination is largely intact in *psm3-K1015R,K106R* mutants is misleading-there are clear defects. I suggest "*psm3-K105R,K106R* mutants retained substantial recombination" as alternative wording.

According to the suggestion, we revised the sentence (line 128).

5. Figure S2 needs a Y-axis label.

According to the suggestion, we add the label.

Reviewer #2 (Comments to the Authors (Required)):

LSA-2024-02606-T

The paper by Li et al described the characterization of non-canonical lysine acetylation of fission yeast Smc3 (Psm3) protein, a component of the cohesin which is critical for sister chromatid cohesion as well as chromosome-axis formation in both mitotic and meiotic cells. The authors identified the acetylation of K1013 and K207 as well as known K105 and K106 in meiosis-specific cohesin complex with Rec8 as alpha-kleisin and Psc3. By acetylation-defective *psm3* mutants, the authors nicely showed that K1013 acetylation plays a redundant role with K105/K106 canonical double-acetylation in chromosome segregation in meiosis I such as monopolar segregation and sister chromatid cohesions (at peri-centromere). The experiments were performed properly and most of the results are technically solid. However, the paper does not provide a molecular insight on the how the loss of Psm3 acetylation leads to cohesin defects in meiosis. It is essential to provide more experimental results to show the molecular mechanism: first, how Moa1 affects the acetylation of Psm3; second, how acetylation of Psm3 promotes mono-orientation of sister-kinetochores; third, how Rec8-cohesin is dissociated from meiotic chromosomes.

We address these concerns in the following points.

Please show page numbers and line numbers. Without these numbers, it is very hard to comment this paper.

According to the suggestion, we added page and line numbers.

Major points:

1. The authors analyzed the role of Psm3 acetylation indirectly by looking at the chromosome

segregation. More importantly, the authors have to show the localization of Rec8-cohesin by immuno-localization of Rec8 protein and chromatin immunoprecipitation of Rec8 at various loci including centromere and sub-centromere (and arm) loci.

We have previously shown that acetylation mutant *psm3-K105R K106R* and *eso1-H17* show no defects in Rec8 cohesin localization (Kagami et al. *EMBO rep* 2011, Fig S5).

2. Given the function of Moa1 in Psm3 acetylation, it is important to check the localization of Moa1 in various acetylation-defective *psm3* mutants.

Similarly, above results (Kagami et al. *EMBO rep* 2011, Fig S5) show that the localization of Moa1 is intact in *psm3-K105R K106R* and *eso1-H17* cells in meiosis I.

3. To know the mechanism of cohesion loss in late prophase I-arrested cells (shown in Figure 4), it is important to know the effect of the *wpl1* mutation in the cohesion loss in various acetylation-defective mutants.

In Fig 3D, we showed that meiotic chromosome segregation defects in acetylation-defective mutants were totally suppressed by *wpl1Δ*, indicating that overall cohesion loss was suppressed. Therefore, we think cohesion assay is not essentially required in this case.

4. Figure 3C and D, Figure 4B and C, graphs: Please show the P-values for the statistical comparison and describe the method in the legend. Without the statistical comparison, it is hard to draw the conclusion.

According to the suggestion, we show the P-values for the statistical comparison in figures (Figure 3C and D, Figure 4B and C).

Minor points:

1. To check the acetylation-defective mutations did not affect the stability of Psm3 protein (and also Rec8), it is very important to analyze the protein level in meiotic cells by western blotting.

As addressed in the comment 1, protein stability and localization are not the main issue of the acetylation-defective mutants. Therefore, we think protein analysis is not essential in this case.

2. Figure 1B: This schematic Figure is misleading since the cell used in the two experiments (in Figures 1C and D) has only one GFP marker on the chromosome 2 (rather than two GFP loci on the single chromosomes). The misleading schematic figure is seen in Figure 4A, in which a single yeast cell has two GFP loci in different chromosome: The results in Figures 4B and C were from a cell with single-GFP locus.

According to the suggestion, we revised schematic figures in Fig 2B and Fig 4A to prevent the misleading.

3. Figure 1C: How the authors quantify the level of acetylation?

The Fig 1C legend quotes Fig S1 which shows the data used for quantification of the level of acetylation.

- How did the authors normalize the values of two strains?
- **We assume that the data are comparable between *wt* and *moa1Δ* cells because similar amounts of proteins were treated in parallel. Indeed, the average difference between *wt* and *moa1Δ* (acetylation in all cohesin subunits, n = 27 sites) is only -0.06 (log2), indicating that acetylation detection levels between two samples are nearly equal. This is shown in the new dot plot graph (Fig S1B).**
- What does Y-axis mean?
- **The Y-axis means the acetylation detection levels explained in Fig S1.**
- What is the reproducibility of this result?
- **We have not repeated this experiment. However, because many acetylation (including canonical acetylation at K105 K106) were detected commonly between *wt* and *moa1Δ* cells (Fig S1), we reasoned that the mass spectrometry analysis and its quantitative data is reliable.**

4. Figure 1E: Please align the amino acid sequences as well the number and marks properly.

Accordingly, we corrected this error.

5. Figure 2C and D, graphs: Please show the raw data by the dot plot, rather than the bar graph.

Accordingly, we show the dot plot in Figure 2C and D.

6. Figure 4C (and D): In Figure 4D, the result for only the *eso1 wpl1* double mutant is shown without that for the *eso1* single mutant.

Figure 3C (and 3D)? The meiotic defects of *eso1Δ* must be more severe than *psm3-K105R K106R K1013R* but not measurable because *eso1Δ* is lethal. However, the meiotic chromosome segregation defects of *eso1Δ* was indeed suppressed by *wpl1Δ* (Fig 4D).

7. Supplemental table: What kind of information in it? Please provide more words to explain the number here.

We now revise the Fig S1 and provide more explanation.

Reviewer #3 (Comments to the Authors (Required)):

This manuscript investigates the regulation of cohesin acetylation and its role in chromosome segregation during meiosis using fission yeast as a model. The authors first biochemically identify the acetylation of Psm3 within the Rec8 cohesin complex. Some of these acetylations are dependent on Moa1, a meiosis-specific centromere protein required for sister kinetochore mono-orientation in meiosis. They then show that these Psm3 acetylations contribute to chromosome cohesion in general during mitosis. Furthermore, Psm3 mutants at their acetylation sites show defects in mono-orientation during meiosis, similar to *moa1D* cells, suggesting that these acetylations are necessary for mono-orientation. However, the Psm3 acetylation site mutants show more severe mono-orientation defects than *moa1D* mutants, more resembling the phenotype of *rec8D* cells. Combining these results with their previous findings that the deacetylase *clr6* mutant partially suppresses the *moa1D* phenotype, the authors suggest that the role of Moa1 is to promote centromere cohesion by antagonizing the deacetylase Clr6 at the centromere, thereby facilitating mono-orientation.

Overall, this manuscript is well constructed with robust experiments and high quality data, and presents logical and clear arguments. The conclusions are strongly supported by the data and there is no overstatement. Their findings include important advancement in molecular understanding of how meiosis-specific chromosome segregation is regulated. Below are some minor suggestions for possible improvements. I recommend this manuscript for publication.

- I agree with the authors that Moa1 antagonizes Clr6 at the centromere. However, a question remains as to how Clr6 promotes cohesin inactivation at the centromere more than at the arms in the *moa1D* mutant (this is the impression I got from the model shown in Fig. 5). Is the activity of Clr6 itself higher at the centromere than at the arms? Or does Clr6 have uniform activity along chromosomes, but during meiosis, Moa1 specifically excludes Clr6's action from the centromere? It would be beneficial if the authors could discuss these possibilities.

We appreciate this suggestion. We now rewrote the text : “We previously showed that the deacetylase mutant *clr6-1* partly suppresses mono-orientation defects in *moa1Δ* cells (Kagami et al., 2011). Importantly, *moa1Δ* cells show cohesion defects only at the core centromere but not along chromosome arms (Yokobayashi and Watanabe, 2005). Therefore, a possible scenario is that Clr6 activity might be high only at the core centromeres and Moa1 would antagonize the Clr6 activity to promote core centromere cohesion and mono-orientation (Fig 5).”. We also revised Fig 5 to prevent mis-impression.

- Figure legend for Fig. 5 is missing.

Accordingly, we now describe the figure legend.

We hope that these changes are satisfactory and that the revised manuscript is now acceptable.

Sincerely,

Yoshinori Watanabe, PhD

March 27, 2024

RE: Life Science Alliance Manuscript #LSA-2024-02606-TR

Prof. Yoshinori Watanabe
Jiangnan University
Science Center for Future Foods
1800 Lihu Avenue
Wuxi, Jiangsu 214122
China

Dear Dr. Watanabe,

Thank you for submitting your revised manuscript entitled "Acetylation of Rec8 cohesin complexes regulates reductional chromosome segregation in meiosis". We would be happy to publish your paper in Life Science Alliance pending final revisions necessary to meet our formatting guidelines.

- please be sure that the authorship listing and order is correct
- please upload your main and supplementary figures as single files
- please add the Twitter handle of your host institute/organization as well as your own or/and one of the authors in our system
- there is a callout for Fig. S1C in the manuscript text, and the corresponding figure has no C panel -- please correct
- please add callouts for Figure S2A-B to your main manuscript text

A. FINAL FILES:

B. MANUSCRIPT ORGANIZATION AND FORMATTING:

**Submission of a paper that does not conform to Life Science Alliance guidelines will delay the acceptance of your

manuscript.**

The license to publish form must be signed before your manuscript can be sent to production. A link to the electronic license to publish form will be available to the corresponding author only. Please take a moment to check your funder requirements.

Sincerely,

Reviewer #1 (Comments to the Authors (Required)):

The authors have addressed my concerns to my satisfaction and I support publication.

March 28, 2024

RE: Life Science Alliance Manuscript #LSA-2024-02606-TRR

Prof. Yoshinori Watanabe
Jiangnan University
Science Center for Future Foods
1800 Lihu Avenue
Wuxi, Jiangsu 214122
China

Dear Dr. Watanabe,

Thank you for submitting your Research Article entitled "Acetylation of Rec8 cohesin complexes regulates reductional chromosome segregation in meiosis". It is a pleasure to let you know that your manuscript is now accepted for publication in Life Science Alliance. Congratulations on this interesting work.

DISTRIBUTION OF MATERIALS:

Again, congratulations on a very nice paper. I hope you found the review process to be constructive and are pleased with how the manuscript was handled editorially. We look forward to future exciting submissions from your lab.

Sincerely,
